# Vaccination in Patients with Liver Cirrhosis: A Neglected Topic

**DOI:** 10.3390/vaccines12070715

**Published:** 2024-06-27

**Authors:** Tommaso Stroffolini, Giacomo Stroffolini

**Affiliations:** 1Department of Tropical and Infectious Diseases, Policlinico Umberto I, 00161 Rome, Italy; tommaso.stroffolini@hotmail.it; 2Department of Infectious-Tropical Diseases and Microbiology, IRCCS Sacro Cuore Don Calabria Hospital, Via Don A. Sempreboni, 5, 37024 Verona, Italy

**Keywords:** cirrhosis, chronic liver disease, vaccines

## Abstract

Patients with liver cirrhosis, due to their weakened innate and adaptive immunity, are more prone to frequent and severe vaccine-preventable infections. Moreover, impaired adaptive immunity results in a limited antibody response to vaccines. Despite this suboptimal antibody response, vaccines have proven to be very effective in reducing severe outcomes and deaths in these patients. In the Western world, regulatory authorities and scientific liver societies (e.g., AASLD and EASL) have recommended vaccinations for cirrhotic patients. However, despite these strong recommendations, vaccine coverage remains suboptimal. Improving vaccine effectiveness and safety information, providing comprehensive counseling to patients, fact-checking to combat fake news and disinformation and removing barriers to vaccination for disadvantaged individuals may help overcome the low coverage rate. In view of this, vaccines should be administered early in the course of chronic liver diseases, as their efficacy declines with the increasing severity of the disease.

## 1. Introduction

Liver cirrhosis patients are immunosuppressed due to deficits in innate (decreased complement activity, reduced chemotaxis, and phagocytosis) and adaptive immunity (decreased memory cells, CD4+ T helper cells and T cell exhaustion) [1,2]. The detrimental effects of cirrhosis on the immune system are collectively defined as cirrhosis-associated immune dysfunction (CAID) [1]. Compared to the general population, individuals with cirrhosis are more likely to contract infections and have a more severe disease course, especially in cases of advanced decompensated cirrhosis [3,4,5]. Of particular relevance is the role of bacterial translocation from the intestinal lumen to mesenteric lymph nodes, which can increase the levels of endotoxins and cytokines [6]. Furthermore, the presence of portosystemic shunting in cirrhotic patients hampers the hepatic clearance of bacteria and endotoxins, increasing the risk of sepsis, multiorgan failure and death [7]. Consequently, the mortality rate for superimposed bacterial infections is nearly four-fold higher in cirrhotic patients than in healthy subjects (38% vs. 10%). Defects in adaptive immunity also generate a limited immune response to vaccines. This well-known hyporesponsiveness to vaccines in this group of patients has led to hesitancy and concern regarding vaccine acceptance and, thus, coverage. The following paragraphs demonstrate that the immune response to vaccines in cirrhotic patients is better than previously supposed, as evidenced by the case of the flu vaccine.

Recent reviews have provided updated information on vaccination in liver cirrhosis [8,9,10]. This review focuses on the impact of certain infectious diseases, such as *S. pneumoniae*, hepatitis A virus (HAV), hepatitis B virus (HBV), flu and Corona Virus Disease 19 (COVID-19), in patients with liver cirrhosis and the immunity coverage and effectiveness of the vaccines against these diseases.

## 2. Streptococcus Pneumoniae

### 2.1. Impact on the Infection 

Bacterial community-acquired pneumonia (CAP) requiring hospitalization is predominantly caused by *S. pneumoniae* in both the general [11] and cirrhotic population [12]. A Spanish study showed that subjects with chronic liver disease (CLD) had a 56.3 times (95% CI = 49.1–64.6) higher risk of hospitalization than age-matched patients without CLD [13]. The risk of a severe course of CAP caused by *S. pneumoniae* is higher in cirrhotic patients compared to non-cirrhotic patients, with increased rates of severe infections (74% vs. 58%; *p* < 0.01), bacteremia (22% vs. 13%; *p* < 0.03) and death (14.4% vs. 7.4%; *p* < 0.03) [12]. Moreover, *S. pneumoniae* is a common cause of spontaneous bacterial peritonitis in patients with cirrhosis, which is not observed in the general population due to immune reasons [14]. The adjusted hazard ratio of death at 30 and 90 days is 2.95 and 2.57, respectively [4]. Overall, the relative risk of death from pneumococcal infection is 5.8 times higher (95% CI = 3.7–9.2) in subjects with cirrhosis than in those without this condition [15]. These severe outcomes are related to the presence of portosystemic shunting in cirrhotic patients, which hampers the hepatic clearance of bacteremia and endotoxins [7]. Additionally, defects in both early and late neutrophil-mediated killing of the microorganism play a significant role in the increased mortality from pneumococcal pneumonia in severely cirrhotic hosts [16]. These findings underscore the necessity for vaccination against *S. pneumoniae* in cirrhotic patients.

### 2.2. The Vaccine

Currently, two vaccines against *S. pneumoniae* are available for adults: a 13-valent conjugate vaccine (PCV13 or Prevenar 13, Pfizer, New York, NY, USA) and a 23-valent polysaccharide vaccine (PPSV23 or Pneumovax 23, Merck, Rahway, NJ, USA). Patients with underlying CLD aged 19–64 years should receive one dose of PPSV23; those aged > 64 years should receive one dose of PPSV23 at least one year after PCV13 or five years after the previous dose of PPSV23 (Table 1).

Data on the immunogenicity of the vaccine are scarce and outdated (more than 20 years old) [17]. Findings have shown a suboptimal antibody response after PPSV23 immunization; antibody levels increased at one month but largely declined at six months in 45 patients with cirrhosis compared to 13 healthy controls [17].

The safety profile is good, as no specific adverse events have been reported in patients with liver cirrhosis following vaccination [18]. Findings on vaccine coverage from the USA and Italy show suboptimal results (Table 2).

Subjects younger than 65 years of age (O.R. 1.7; 95% CI = 1.01–2.87) and those not married (O.R. 2.10; 95% CI = 1.26–3.56) were more likely to be vaccinated in an American study [19]. A lack of vaccination was more likely reported in subjects under 65 years of age (O.R. 3.39; 95% CI = 2.41–4.76) and in those with alcoholic cirrhosis (O.R. 1.91; 95% CI = 1.03–3.52) in an Italian study [22]. In the latter study, as many as 91.7% of unvaccinated subjects reported a lack of information as the reason for their vaccination status, suggesting the need for better counseling from treating physicians to overcome misinformation.

## 3. Flu

### 3.1. The Impact of the Infection

Influenza is one of the most common infectious diseases worldwide, able to infect nearly 20% of the global population and responsible for nearly 600,000 deaths each year [8]. In cirrhotic patients, flu may have a relevant impact in terms of morbidity and mortality. During the years 2013/2014, a survey in 19 hospitals located in Russia, Turkey, China and Spain showed a risk of hospitalization nearly 2-fold higher in subjects with cirrhosis [23]. In 2009, the pandemic of influenza A (H1N1) was associated with a more than 5-times increased risk of hospitalization and a 17-fold increased risk of death in cirrhotic patients [24]. Other authors have reported a risk of death associated with Influenza A (H1N1) in those patients that was 3–4 times higher than in subjects without cirrhosis [25,26]. In a more recent study during the 2016–2018 season, the adjusted hazard ratio for mortality in patients with cirrhosis was 3.94 (C.I. 95%= 1.07–14.45) [27]. Clinical presentation of flu in cirrhotic patients may be acute liver decompensation [28] or acute on chronic liver failure (ACLF) [29]. 

### 3.2. The Vaccine

Influenza vaccines are designed several months before the flu season, containing dominant strains from the WHO global influenza surveillance. Two inactivated vaccines are available for cirrhotic patients: the trivalent (two strains of Influenza A and one strain of influenza B) and the quadrivalent (two strains of influenza A and two strains of influenza B). Vaccine administration is recommended yearly. The immunogenicity of the flu vaccine results much better than it was supposed. The seroconversion rates are 80% for A/H1N1 and 87% for the B strain in patients with CLD [30]. It results in 75% in patients with HBV-related cirrhosis and 85% in hepatitis C virus (HCV)-related cirrhosis, respectively [31]. Despite these reassuring findings, the vaccine coverage results are limited to subjects with CLD, mostly because primary and secondary care physicians are concerned for the potential of flu vaccines both in providing immunogenicity and effectiveness against the health complications of flu in subjects with CLD [32]. In Alabama, at the beginning of this century, flu vaccine coverage was 55% [21]; in Minnesota, coverage increased from 36.1% in 2007 to 65.8% in 2015 [20]. The U.S. National Health Interview Survey has shown that 55% of CLD patients had been vaccinated in 2016 [33]. During the 2015/2016 season, only 42% of patients with CLD had been vaccinated in the UK [34]. In Italy, the overall coverage in 2019 was 39.6% but was only 26.9% in patients younger than 65 years of age, among patients with liver cirrhosis of any etiology [35]. (Table 3).

In the latter survey, patients younger than 65 years of age (O.R. 2.8; 95% CI = 1.68–3.36), people from abroad (O.R. 2.7; 95% CI = 1.1–4.0) and those with an alcoholic etiology for cirrhosis (O.R. 2.4; 95% CI = 1.5–3.8) were all identified as groups at increased risk of a lack of vaccination, and, therefore, they require particular attention. Furthermore, in this survey, as many as 71.4% of unvaccinated subjects reported a lack of information from their physicians as the reason for their vaccination status. A previous Italian survey in 2005 showed that patients with cirrhosis were less likely to be vaccinated than those affected by other CLD, confirming that liver cirrhosis represents a barrier to flu vaccination [36]. The vaccine has been shown to be effective against flu-related morbidity. Vaccinated CLD patients were 27% less likely to be hospitalized than the unvaccinated (risk ratio 0.73; 95% CI = 0.66–0.80) [30]. An Asian controlled trial of 311 cirrhotic patients showed that the rates of culture-positive influenza and influenza-related complications were lower in vaccinated patients compared to unvaccinated patients [37]. Finally, no safety concerns have emerged from the flu vaccine in cirrhotic patients [38].

## 4. HAV

### 4.1. The Impact of the Infection

It has been well known for several years that acute HAV in patients with CLD has a great impact in terms of mortality. During the years 1983–1987, among the 115,000 reported acute HAV cases, 28% of the 381 observed deaths occurred in patients with CLD [37]. In 1998, during the acute HAV Shanghai epidemic involving 300,000 subjects, the case fatality rate was 5.6-fold higher in subjects with underlying CLD [38]. Fatal outcomes of acute HAV have been reported in patients with CLD of any etiology, including 35% of deaths in HCV-related CLD [39], a 55% rate of fulminant hepatitis in chronic HBsAg carriers of HBV [40] and high rates in alcohol-related liver cirrhosis [41].

### 4.2. The Vaccine

Two vaccines are currently available: HAVRIX and VAQTA. Two doses at 0 and 6 months are recommended. Regulatory authorities in the USA and several European countries recommend the HAV vaccine for CLD patients, as a large proportion of adults are still naïve to the viral infection. In contrast, in India, where the prevalence of people with CLD who have already been exposed to HAV is >90%, vaccination is not recommended [42].

The immunogenicity of the vaccine is quite satisfactory. Antibody titers are 94–98% in patients with a not-advanced stage of CLD [43]. The response decreases in compensated cirrhosis to 71% for patients at the CHILD B stage and to 57% for those at the CHILD C stage [44], supporting the need to administer the vaccine at an early stage of CLD [32]. An accelerated three-dose (0, 1, 2 months) regimen has shown a 23% improved response rate [45]. Vaccine coverage shows very low rates (Table 4).

The vaccination rate was 26% in Alabama in 2001, with a similar rate of 26% found in the National Health and Nutrition Examination Survey (2005–2008) [46]. In addition, a rate of 20.7% was reported in the US Veterans Affairs database [47], 7.7% was reported in Cleveland during the years 2004–2013 [19], 26.4% was reported in the Minnesota study during the period of 2007–2015 [20], and 14.6% was reported in France in 2017. Real-world data have confirmed the efficacy of the vaccine: acute HAV infection in patients with underlying CLD occurs significantly more in unvaccinated individuals than in vaccinated subjects (0.16% vs. 0.01%) with an incidence ratio of 14.25 (95% C.I. = 2.2–59.5) [47]. Last, the vaccine’s safety has been extensively documented, with no reported serious adverse events after vaccination in CLD patients [43,44].

## 5. HBV

### 5.1. The Impact of the Infection

Acute hepatitis B in patients with preexisting CLD may lead to liver failure and death [47,49]. Accelerated progression to decompensated cirrhosis and HCC have been linked to HBV co-infection. Most cases occur against the background of HCV-related CLD, worsening the disease course in nearly 30% of patients [50] and increasing the risk of HCC development [51].

### 5.2. The Vaccine

Recombinant DNA vaccines are available: EngerixB, Recombivax HB (administered in three doses at 0, 1 and 6 months for both) and HEPLISAV B (administered in two doses 4 weeks apart). The immunogenicity of the vaccine is lower in patients with CLD compared to immunocompetent adults [52]. Moreover, it declines according to the severity of CLD, being 88% in CHILD B patients [53,54] and as low as 16–20% for CHILD C patients [52]. Attempts to increase immunogenicity in advanced cirrhosis using increased dosing [52,53] and an accelerated administration schedule [55,56] have resulted in suboptimal outcomes. Antibody responses may vary according to the etiology of CLD, resulting in 44–75% in the alcohol-related form [54,57] and 69–100% in the HCV-related form [58,59]. Patients with NAFLD-related CLD without cirrhosis have an unimpaired response [60]. It has been recently shown that HCV-positive subjects, previously non-responders to HBV vaccines, developed a proper response to the vaccine in 48% of cases once revaccinated after HCV eradication following DAAs [61]. It is also well described [62] that the immune response rate to HB vaccination is inversely related to the age of the subjects, decreasing in those with older ages. This represents a further valuable argument to vaccinate patients with CLD at an early stage of disease progression, as patients with liver cirrhosis are older than those with early-stage CLD. Vaccine coverage is very low: from 11.0% [19] to 32.1% [46] in US-based studies, to 36% in France [48], 24% in Germany [63] and 16.3% in a multicenter cross-sectional study in Italy enrolling 731 cirrhotic cases of any etiology [64] (Table 5).

Poor knowledge and unjustified concerns from primary care physicians [65] and a lack of motivation among hepatologists [66] have been identified as reasons for low vaccine coverage in the US. In another Italian study [64], unvaccinated patients reported a lack of information from treating physicians in 78.5% of cases as the reason for their vaccination status. Finally, the HB vaccine is safe in patients with CLD [52,53,54].

## 6. SARS CoV-2 

### 6.1. The Impact of the Infection

Several waves of COVID-19 infections associated with emerging variants have occurred. The first wave started at the beginning of 2020 due to the spread of the Wuhan Hu1 wild type. A second wave due to the Alpha variant emerged in autumn 2020, and in spring 2021, the Delta variant emerged, characterized by a higher risk of both transmissibility [67] and severity [68]. In autumn 2021, the Omicron variant emerged, showing reduced pathogenicity [69] but further increased transmissibility [70] and higher potential for evading both natural and induced immune responses [71]. Finally, the Eris and Pirola variants, both subvariants of Omicron, emerged in summer and autumn 2023. These were further characterized by decreased pathogenicity and increased transmissibility (Table 6).

Data on the impact of COVID-19 infections in patients with CLD mostly come from the former circulating strains of the pandemic virus (Alpha, Delta). Patients with CLD, particularly those with cirrhosis, have impaired immune function, so they have limited potential for clearing SARS-CoV-2. Indeed, disease progression and death are higher in this population [72]. Patients with cirrhosis, once infected with COVID-19, may develop a severe course for both diseases, including a more severe COVID-19 illness and worsening of the pre-existing liver disease, with a higher and more prolonged rate of hospitalization, liver decompensation and death [8].

A report of two combined international liver registries (SECURE-Cirrhosis and COVID-Hep.net) evaluated 745 CLD patients (of whom 386 had cirrhosis), showing a mortality rate that was four-fold higher in cirrhotic CLD COVID-19 infected patients compared to non-cirrhotic CLD patients (32% vs. 8%) [73]. Further evidence is provided by the National COVID Cohort Collaborative dataset from the US: the hazard risk for mortality at 30 days in subjects with cirrhosis and COVID-19 infection was 2.4 times (95% C.I. = 2.2–2.6) higher than in cirrhotic patients without COVID-19 infection and 3.3 times (95% C.I. = 2.9–3.8) higher than in those with COVID-19 infection without cirrhosis [74].

### 6.2. The Vaccine

Several COVID-19 vaccines were approved between 2020 and 2023. These include mRNA vaccines (Pfizer, New York, NY, USA; Moderna, Cambridge, MA, USA), an adjuvant recombinant protein vaccine (Novavax, Gaithersburg, MD, USA) and vaccines that use a replication incompetent adenovirus vector (Johnson & Johnson, New Brunswick, NJ, USA, Oxford-AstraZeneca). At the end of 2020, vaccination against COVID-19 started with a favorable impact on COVID-19-related morbidity and mortality [75,76]. Vaccines have been adapted according to the emerging variants. In the Western world, liver societies (AASLD and EASL) recommend vaccination against COVID-19 in CLD and cirrhosis patients to prevent a severe course and reduce mortality in these patients [77,78]. Despite a low degree of pathogenicity in immunocompetent subjects, the variants currently circulating may lead to severe outcomes in immunodeficient subjects, including cirrhotic patients. Indeed, vaccination continues to be advised in these patients. This recommendation is supported by findings from a recent USA study enrolling 60,448 hospitalized patients for COVID-19 during the period of June 2021–March 2023. Despite the proportion of subjects with critical outcomes declining from 24.8% to 19.4% between the Delta and the Omicron variants, the risk of a critical outcome in unvaccinated patients across periods continued to be associated with the presence of comorbidities (adjusted RR 2.27; C.I. 95% = 2.14–2.41) [79].

An impaired vaccine response (76%) was found in patients with CLD [80]. However, no significant immunogenic difference has been found according to the severity of CLD, with response rates of 77% in non-cirrhotic CLD patients, 79% in compensated cirrhotic patients and 77% in decompensated cirrhotic patients [81]. Few surveys have evaluated vaccine coverage against COVID-19 in cirrhotics. In the USA, the rate of coverage was approximately 60% in cirrhotic patients regardless of the disease stage [82]; in China, a rate of 31.7% among decompensated cirrhosis patients was found [83]; and in Italy, a higher rate of 89.7% among cirrhotics of any stage (91% in CHILD A and 83.8% in CHILD B-C) was observed [84] (Table 7).

In the above-mentioned Italian study, as many as 86.4% (131/154) of unvaccinated subjects reported “refusal” as the cause of not being vaccinated. A relevant proportion of 10.4% unvaccinated subjects reported having received negative advice toward vaccination from their treating physician [84]. 

The impact of vaccination on the course of infection according to different variants was recently evaluated [75]. In cases of infection due to the Alpha and Delta variants, vaccinated individuals had a milder disease course, a shorter duration of positivity to the virus and a lower viral load compared to unvaccinated subjects. Conversely, these positive effects in vaccinated people have been attenuated since the emergence of the Omicron variant. In a cohort study of USA veterans with cirrhosis, the administration of two vaccine doses showed an overall decreased risk of death compared to unvaccinated subjects (adj HR 0.21; 95% C.I. = 0.10–0.42), which was lower in compensated cirrhosis (adj HR 0.19; C.I. = 0.08–0.45) compared to decompensated cirrhosis (adj HR 0.27; 95% C.I. = 0.08–0.90) [85]. However, breakthrough COVID-19 infections were observed despite vaccination. To overcome breakthrough infections, the same group of authors administered a third dose of the vaccine in cirrhotic patients. This policy showed an 80.7% reduction in breakthrough infections, a 100% reduction in severe/critical disease course and a 100% reduction in death, suggesting that the third dose overcomes vaccine hyporesponsiveness [86]. Similar results were observed in an Italian study among 1358 vaccinated cirrhotic patients of any etiology; breakthrough infections occurred less frequently in those who received three doses of the vaccine than in those who received two doses (9% vs. 33.9%, *p* < 0.01) [84]. An analysis of 168,245 vaccinated patients with cirrhosis, aimed at removing vaccination hesitancy/resistance caused by concerns for safety, showed reassuring evidence for the safety profile of vaccination [76].

Finally, vaccines against COVID-19 have been the subject of new and unexpected challenges. Despite the change in the course of the disease brought about by vaccines, with an impressive reduction in morbidity and mortality, certain groups have refused evidence-based data due to following fake news. Most importantly, these groups have adopted aggressive behaviors and conduct against eminent scientists, even resorting to threats of death [87]. Conversely, and very welcomed, the COVID-19 pandemic has accelerated vaccine research. Most of the rapidly approved vaccines were deployed using novel platforms, such as mRNA techniques; these platforms may be adopted in other disease areas.

## 7. Conclusions

Vaccines generally elicit a suboptimal antibody response in patients with CLD due to an impaired immune system. However, this immune dysfunction also contributes to a more severe course of various infectious agents in cirrhotic patients compared to the general population. Despite the impaired immune response to vaccination, vaccines have proven effective in preventing negative outcomes of infections in cirrhotic patients, providing both cost and overall health benefits.

Awareness of the limited immunogenic power of vaccines in cirrhotic patients has led to concern and hesitancy among physicians, resulting in largely suboptimal vaccine coverage in these patients. Additionally, a discrepancy in vaccination practices has been observed between general practitioners (GPs) and specialists, with flu and pneumococcal vaccination being more commonly provided by the former group, whereas HAV and HBV vaccinations are more frequently administered by the latter group [18].

Several interventions may be useful in addressing the disappointing vaccine uptake (Table 8).

Informative campaigns targeting healthcare providers are essential to educate them about the availability, effectiveness, safety and risk–benefit balance of vaccines in CLD patients. The rebuttal of fake news spread by some media and online social platforms is crucial, as it has led to hesitancy even among healthy individuals, as witnessed during the COVID-19 pandemic. An Italian study revealed that unvaccinated cirrhotic patients refused anti-COVID-19 vaccination due to hesitancy stemming from personal or political beliefs in as many as 86.4% of cases, and 10% of cases received negative advice on vaccination from their GPs [84]. Moreover, individuals with fewer years of schooling were found to be nearly twice (CI95% = 1.2–2.9) as likely to be unvaccinated compared to those with higher education levels [84].

Several studies have highlighted the most frequent cause of an unvaccinated status among patients as a lack of information from healthcare providers [23,35,63,64,65]. Therefore, informative counseling by physicians with patients is paramount to counter disinformation, misinformation and hesitancy. Electronic medical record reminders (EMRs) sent to GPs could serve as a useful tool to improve vaccine coverage, as evidenced by a 12.5% increase in pneumococcal vaccine coverage following the adoption of reminders to clinicians by EMRs [88].

Barriers to accessing healthcare for disadvantaged populations such as migrants, the homeless and those from low social classes present challenges in some high-income countries [89]. These barriers are often related to a low income, isolation and a lack of insurance coverage [90]. Another crucial consideration is the decreasing immune responsiveness to vaccines in CLD patients as the severity of liver disease increases, compounded by older age. Hence, administering vaccines early in the course of the disease is imperative to achieve a better immunological response.

Last, the COVID-19 era has ushered in new opportunities for advancements in vaccine technologies, potentially leading to improvements in vaccine development for other communicable diseases.

## Figures and Tables

**Table 1 vaccines-12-00715-t001:** Recommended pneumococcal vaccination schedule in patients with CLD. Adapted from Reference [8].

Patients 19–64 years old	1 dose of PPSV23
Patients > 64 years old	1 dose of PPSV23 at least 1 to 5 years after PCV13 or PPSV23, respectively

**Table 2 vaccines-12-00715-t002:** Vaccination coverage against *S. pneumoniae* in cirrhotic patients.

Coverage (%)	Country	Year	Ref.
19 (specialist center); 39.1 (primary care)	USA	2003	[18]
19.9	USA	2004–2013	[19]
63	USA	2007–2015	[20]
34	USA	2001	[21]
17.9	ITALY	2022	[22]

**Table 3 vaccines-12-00715-t003:** Vaccination coverage against Flu in cirrhotics.

Coverage (%)	Country	Year	Ref.
55	USA	2001	[21]
36.1–65.8	USA	2007–2015	[20]
55	USA	2016	[33]
42	UK	2015–2016	[34]
39.6	Italy	2019	[35]

**Table 4 vaccines-12-00715-t004:** Vaccination coverage against HAV in cirrhotics.

Coverage (%)	Country	Year	Ref.
26	USA	2001	[21]
26	USA	2005–2008	[46]
20.7	USA	2010	[47]
7.7	USA	2004–2013	[19]
26.4	USA	2007–2015	[20]
14.6	France	2017	[48]

**Table 5 vaccines-12-00715-t005:** Vaccination campaign against HBV in cirrhotics.

Coverage (%)	Country	Year	Ref.
26	USA	2001	[21]
32.1	USA	2005–2008	[46]
21.9	USA	2010	[47]
11	USA	2004–2013	[19]
24.7	USA	2007–2015	[20]
36	France	2017	[48]
24	Germany	2019	[63]
16.3	Italy	2019	[64]

**Table 6 vaccines-12-00715-t006:** Temporal trends of COVID-19 variants.

Variant	Period
Wuhan Hu 1	Beginning 2020
Alpha	Autumn 2020
Delta	Spring 2021
Omicron	Autumn 2021
Eris (Omicron subvariant)	Summer 2023
Pirola (Omicron subvariant)	Autumn 2023

**Table 7 vaccines-12-00715-t007:** Vaccination coverage against COVID-19 in cirrhotics.

Coverage (%)	Country	Year	Ref.
60	USA	2021	[82]
37.1	China	2021	[83]
89.7	Italy	2021	[84]

**Table 8 vaccines-12-00715-t008:** Interventions to improve vaccine coverage in adults.

Informative campaigns to increase vaccine awareness in healthcare providers
Rebuttal of fake news associated with vaccines
Informative counseling with patients
Electronic medical record reminders (EMRs)
Removal of barriers to access vaccination for disadvantaged people

## Data Availability

Not applicable.

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
