# Peer review of "Vaccination in Patients with Liver Cirrhosis: A Neglected Topic"

_vaccines, 2024, doi:10.3390/vaccines12070715_

Round 1
Reviewer 1 Report
Comments and Suggestions for Authors
Please see all suggested corrections/modifications, highlighted in the attached manuscript file.

Moderate language polishing required.
Author Response
Reviewer 1
We appreciate your thorough assessment of our manuscript. We have revised the text according to your suggestions, and these changes are now highlighted in the document.
Regarding your comments:
- “Of the hepatitis viruses, HBV, HCV, HDV…. Whereas HAV is not known to cause chronic liver infection or cirrhosis.” (Line 44)
- “Please cross-check. […] patients with HAV-induced cirrhosis also received HAV vaccine.” (Line 142)
- “HCV has been discussed under subhead HBV?” and “HCV has been discussed under subhead HBV vaccine?” (Lines 177 and 190)
We do not state the points highlighted by your comments in these sections. This may be due to a misunderstanding. Our discussion focuses on the impact of acute HAV and HBV infections and the features of vaccination in patients with pre-existing chronic liver disease (CLD) caused by HCV, HBV, or other causes. So: patients with pre-existing CLD or cirrhosis caused by non HAV infection (clearly, since that agent cannot cause chronic liver infection).
Reviewer 2 Report
Comments and Suggestions for Authors
This manuscript provides a clear pictures of which vaccinations should be considered to be inoculated in patients with cirrhosis, and it is a timely. The field of appropriate and beneficial vaccinations in cirrhotic patients is not extensively investigated, thus resulting in limited information available to hepatologists, and clinicians in general.
The major bacterial and viral pathologies that could affect cirrhotic patients, and the available vaccination are discusses in the manuscript to the extent supported by published studies on the topics.
The manuscript is properly written for the most part. There are a couple of sentences that could benefit of some rewording (see section Comments on the Quality of English Language).
Comments on the Quality of English LanguageThe style of English Language is essentially fine. There are a couple of typos or inaccuracies that need to be amended.
Line 257: "...In the USA, the rate resulted around 60% in cirrhosis of 257 any stage [82]." would read better "...In the USA, the rate of coverage was approximately 60% in cirrhotic patients regardless of the disease stage [82]
Line 260: verb is missing at the end of the sentence: e.g. '...were observed'
Author Response
Reviewer 2
We appreciate your thorough assessment of our manuscript. We have revised the text according to your suggestions, and these changes are now highlighted in the document.
Reviewer 3 Report
Comments and Suggestions for Authors
This review focuses on the impact of certain infectious diseases, such as S. pneumoniae, HAV, HBV, influenza, and COVID-19, in patients with liver cirrhosis and the immunity coverage and effectiveness of vaccines against these diseases. In general, the scientific problem investigated by the authors of this article is relevant, and the results of this study may be useful to the readers of Vaccine Journal. However, I have a number of comments about the results of this paper:
(1) There is no research methodology section indicating the type of review (systematic, narrative, etc.), the databases used, the principles of inclusion and exclusion of information, and the limitations of the study. In particular, there is no mention of whether evidence-based medicine methods were used in the studies cited by the authors.
(2) It is not clear from the review data whether different vaccinations are shown for stage IV cirrhosis (liver failure or advanced liver disease or liver failure).
(3) It is not clear whether the authors of the reviewed publication analyzed the possible adverse effects of vaccinations in patients with cirrhosis or whether this was not part of the study.
(4) A similar study has already been performed by Alukal et al. (see references) as well as in "Airola, C.; Andaloro, S.; Gasbarrini, A.; Ponziani, F.R.. Vaccine responses in patients with liver cirrhosis: From the Immune System to the Gut Microbiota. Vaccines 2024, 12, 349. https://doi.org/10.3390/vaccines12040349". What is the fundamental novelty of this study?
(5) References do not conform to MDPI and ACS style.
Author Response
Reviewer 3
We appreciate your thorough assessment of our manuscript. We have revised the text according to your suggestions, and these changes are now highlighted in the document.
1) This is a narrative review, and it has been written according to “Vaccines” standards and instructions for authors section. In particular: “Review:Reviews offer a comprehensive analysis of the existing literature within a field of study, identifying current gaps or problems. They should be critical and constructive and provide recommendations for future research. No new, unpublished data should be presented. The structure can include an Abstract, Keywords, Introduction, Relevant Sections, Discussion, Conclusions, and Future Directions.“
Similarly, the work by Airola et al. published in the same journal presents a very comparable structure.
2) These data are included whenever specified in the original research. When such data are not available, they are not reported.
3) Data regarding the safety and possible adverse effects of vaccinations are reported in references [18], [38], [43-44], and [52-54], [76] cited in the relevant sections.
4) Our work differs from the studies by Alukal et al. 2021 and Airola et al. 2024 in several significant ways. Our review is designed to provide clinicians with a practical, updated overview of vaccinations, their effectiveness, and the coverage of various vaccines specifically in patients with liver cirrhosis. We emphasize the social and public health implications of these vaccinations, offering actionable insights and recommendations for clinical practice. Additionally, we report comprehensive vaccine coverage data, which is crucial for understanding the extent and effectiveness of immunization efforts in this vulnerable population. Furthermore, our review incorporates very recent studies and the latest research findings, ensuring that the information we provide is up-to-date and relevant to current clinical practice. This allows clinicians to make informed decisions based on the most current evidence available. Similarly, the study by Alukal et al. addresses the efficacy and safety of vaccinations, in particular in special populations, and is updated to 2021. This is one of the reasons we cited the reference. The study by Airola et al. focuses on the immunological aspects of vaccination, particularly the interactions between the immune system and the gut microbiota, which is a more specialized area of research.
5) We appreciate the reviewer's suggestion. The reference format now conforms to the journal style.
Round 2
Reviewer 1 Report
Comments and Suggestions for Authors
Satisfactory response
Reviewer 3 Report
Comments and Suggestions for Authors
The clarifications and corrections made by the authors to my comments are clear, I have no further comments.